# OpenReview forum: "Trojan-Speak: Bypassing Constitutional Classifiers with No Jailbreak Tax via Adversarial Finetuning"
_ICML.cc/2026/Conference — ICML 2026 spotlight_

### Official Review · Reviewer_rTxy · 2026-03-10

**Soundness:** 2
**Presentation:** 3
**Significance:** 2
**Originality:** 3
**Overall Recommendation:** 3
**Confidence:** 4

**Summary:**

This paper introduces Trojan-Speak, an adversarial fine-tuning method that bypasses Anthropic’s Constitutional Classifiers.

**Compliance With Llm Reviewing Policy:**

Affirmed.

**Final Justification:**

believe the novelty of the whole work is not enough to meet my expectations. I will maintain my score.

**Key Questions For Authors:**

See weakness

**Limitations:**

Yes

**Strengths And Weaknesses:**

Strength:

1. This paper proposes a curriculum-based adversarial fine-tuning pipeline combined with hybrid RL+SFT training, which enables models to learn encoded communication while largely preserving their original reasoning capabilities.

2. Extensive empirical evaluation across models, classifiers, and reasoning benchmarks.

Weakness:

1. Limited novelty compared to prior cipher-based jailbreak attacks.

This work relies on encoding harmful content into an alternative token representation so that the classifier cannot recognize the underlying semantics. However, this idea is similar to existing cipher-based jailbreak techniques. The attack creates a distribution shift where the classifier sees encoded tokens while the model learns to interpret them, which explains the improved bypass results rather than introducing a new attack paradigm.

2. Unclear practical deployment scenario of the learned encoding scheme.

Unlike standard ciphers that are widely known, the proposed encoding scheme is custom-designed and learned during fine-tuning. It is therefore unclear how this model would be used in practice. If the model is for the attacker’s own use, it is unclear why such complex training is necessary. On the other hand, if the model is intended to be used by other users, the algorithm should be public, making no difference from other encoding techniques.

3. Potential robustness issues of the encoding-based communication protocol.

Encoding schemes based on character substitutions can be fragile. Small perturbations such as typos, formatting changes, or tokenization differences may affect the decoding process and lead to incorrect results. I suggest the authors evaluate the robustness of the learned encoding under such perturbations.

---

> ### Author Rebuttal · Authors · 2026-03-31
>
> We thank the reviewer for their thoughtful review and suggestions for our work. We were pleased to hear that it is recognized to have "extensive empirical evaluation". We have acted on the reviewer's suggestions on perturbation analysis.
>
>
> # Regarding W1
>
> We appreciate the reviewer's observation and acknowledge that our work shares conceptual roots with the broader family of cipher-based approaches. However, we believe the contribution extends beyond simply creating a distribution shift. The critical distinction of our method is that it achieves high bypass rates *without any capability degradation*, which prior cipher-based and adversarial fine-tuning methods have consistently failed to do. Existing approaches like CMFT and related techniques report significant drops on reasoning benchmarks, often exceeding 25% degradation, because the model sacrifices general ability in order to learn the adversarial behavior. Achieving a high bypass rate alone is not novel; what is novel is demonstrating that the assumed trade-off between bypass rate and capability retention is not fundamental. Our method represents a Pareto improvement over the prior attack landscape, and we believe this is a meaningfully distinct and practically important result even if the surface-level mechanism resembles existing cipher-based techniques.
>
> # Regarding W2
>
> The key insight here is that the encoding scheme is not a single fixed cipher that a defender can learn and block. It is *generated* through the fine-tuning process, and the space of possible encodings is combinatorially vast. Unlike known ciphers such as Base64 or Caesar shifts, where a defender can hardcode detection rules or train against a closed set of transformations, our approach allows an attacker to produce an essentially unbounded number of valid mappings simply by varying the fine-tuning procedure. Defending against this requires anticipating and guarding against all possible encoding schemes, which is a computationally intractable problem. In terms of deployment, the practical threat model is straightforward: an attacker fine-tunes the model with a chosen encoding and then distributes the model (for instance, via an API or open-weight release) alongside a lightweight decoding key shared out-of-band. Any user who knows the encoding can then query the model to produce harmful content that passes through the constitutional classifier undetected, while the defender is left facing an effectively infinite search space of possible encodings to guard against.
>
> # Regarding W3
>
> In order to show the model's robustness against perturbations, we have perturbed the case in the following way.
>
> **Char-sub:** Randomly replaces alphabetic characters with different lowercase letters at probability p
>
> **Transpose:** Randomly swaps adjacent character pairs at probability p
>
> **Whitespace:** Randomly inserts spaces after characters at probability p
>
> Qwen3-32B Performance on GPQA-D (TS: 48.4)
> | Perturbation | 1% | 5% |
> |--------------|------|------|
> | Char-sub | 48.5 | 48.4 |
> | Transpose | 48.1 | 48.6 |
> | Whitespace | 48.8 | 48.6 |
>
> As we see in the table, Trojan-Speak model does not get affected by the perturbations as the RL stage also optimizes the model for these mistakes in encoding/decoding.
>
> ---
>
> # Conclusion
>
> We thank you in advance for checking our rebuttal and reconsidering your score if you believe our explanations and the introduction of new results alleviated your concerns.

---

> > ### Author Rebuttal · Reviewer_rTxy · 2026-04-01
> >
> > I think most of my concerns have been addressed, I will maintain the score.

---

> > > ### Author Response · Authors · 2026-04-03
> > >
> > > Dear Reviewer, we appreciate you taking the time to read our rebuttal. We would love to know if you still have any remaining concerns with the paper. We'd also like to invite you to review the new results introduced in our rebuttals to the other reviewers, as some reviewers have reconsidered their scores in light of them . If you believe your concerns are fully resolved, we would kindly ask that you reconsider your score as well. Thank you!

---

### Official Review · Reviewer_k96T · 2026-03-11

**Soundness:** 2
**Presentation:** 3
**Significance:** 2
**Originality:** 2
**Overall Recommendation:** 4
**Confidence:** 2

**Summary:**

The paper introduces Trojan-Speak, an adversarial fine-tuning method that trains models to communicate harmful information through an encoded protocol that bypasses LLM-based safety classifiers. The approach combines curriculum learning with hybrid RL and supervised fine-tuning to teach the model this encoded communication while preserving most of the model’s original capabilities. Experiments show that fine-tuned models can evade Constitutional Classifiers and provide detailed responses to CBRN queries while maintaining high performance on standard reasoning benchmarks.

**Compliance With Llm Reviewing Policy:**

Affirmed.

**Final Justification:**

Please add HarmBench and AdvBench results to the paper. Other than that, it looks good. Score changed.

**Key Questions For Authors:**

The experiments primarily evaluate Trojan-Speak on CBRN-related queries. Have the authors tested whether the encoded communication protocol generalizes to other safety domains (e.g., cybercrime, fraud, or violence-related content)?

The evaluation focuses mainly on Constitutional Classifiers. How does the attack perform against other types of safeguards, such as generative refusal models, multi-stage moderation pipelines, or ensemble filtering systems?

The paper evaluates a specific model family during fine-tuning. Have the authors tested whether the trained Trojan-Speak protocol transfers across different model architectures or sizes?

**Limitations:**

The paper acknowledges that some implementation details are intentionally withheld and briefly discusses the defensive implications of the attack, but the discussion of limitations and societal impact could be more explicit.

**Strengths And Weaknesses:**

The paper proposes a clearly defined adversarial fine-tuning pipeline that combines curriculum learning with hybrid RL and supervised fine-tuning to train models to communicate using an encoded protocol that bypasses content classifiers. The methodology is described in a structured way, and the evaluation includes experiments demonstrating high classifier evasion rates while maintaining performance on standard reasoning benchmarks. The paper also evaluates the attack against a strong real-world safeguard (Anthropic’s Constitutional Classifiers), which strengthens the empirical relevance of the work. The authors additionally discuss possible defensive directions such as activation-level probes, which shows awareness of the broader security implications.

The paper is generally well organized and provides a clear narrative from motivation to method and evaluation.

The paper addresses an important security concern: whether LLM-based safety classifiers remain effective when adversaries have access to fine-tuning APIs.

The paper introduces a novel combination of techniques for adversarial fine-tuning, particularly the use of curriculum learning to teach models an encoded communication protocol while preserving capability.

Some implementation details are intentionally withheld due to safety concerns, which makes it difficult to fully evaluate reproducibility and verify the reported results. The experiments also focus on a relatively narrow class of harmful queries (CBRN-related content), and it is unclear how broadly the attack generalizes across other safety domains. In addition, the evaluation does not fully explore how robust the attack is to defensive measures such as randomized classifiers, rate limiting, or monitoring systems.

Although the attack highlights a potential weakness of classifier-based safeguards, the threat model assumes that attackers can fine-tune models in ways that may not be allowed or easily achievable in many production environments. The practical impact therefore depends on the specific deployment settings of model providers.

Many components of the approach—such as substitution ciphers, adversarial fine-tuning, and reinforcement-learning-based training—have been explored in prior work. The novelty mainly lies in their combination and in demonstrating that capability degradation can be avoided, rather than introducing fundamentally new attack primitives.

---

> ### Author Rebuttal · Authors · 2026-03-31
>
> We thank the reviewer for taking the time to review our paper, and we are excited to hear that our paper is recognized as addressing "an important security concern". We have acted on their suggestions and introduced new results on out-of-domain harmful queries.
>
> The reviewer's main concerns are testing the protocol on various model families, out-of-domain generalization of Trojan-Speak, and attacks' performance in addition to Constitutional Classifiers.
>
> # Cross-architecture / cross-scale transfer
>
> We believe the reviewer might have missed some of our extensive validation of the training protocol on various model sizes and families. **§4.1, Table 2, Fig 4, Fig 6** evaluate **five models across two architecture families** (Anthropic Claude + Qwen3, 4B–32B), same pipeline, zero architecture-specific tuning, in order to show the effect of model scale on the learnability of Trojan-Speak.
>
> # Generalization beyond CBRN
>
> In order to show the dangers of Trojan-Speak, we decided to choose a difficult CBRN benchmark to not merely show harmful behavior, but also show capability. We have run our same Qwen3-32B model on HarmBench and AdvBench. The model consistently gets >99% ASR with >99% classifier evasion in both benchmarks, highlighting that Trojan-Speak enables models to be harmful in various categories while still bypassing classifiers.
>
> # Threat-model realism
>
> Finetuning APIs are offered by major AI providers (e.g., OpenAI, Google), and our results demonstrate that adversaries with fine-tuning access do not need to trade off between evasion effectiveness and model utility. We encourage the community to develop defenses that operate at multiple levels of abstraction.
>
> # Reproducibility
>
> In the appendix, we fully show all the steps required to recreate our results, including all of the hyperparameters.
>
> ---
>
> We hope we have both clarified and addressed your concerns. We kindly ask that you reconsider your score based on our response. Feel free to let us know if you'd like us to address anything further. Thank you again for your thoughtful feedback!

---

> > ### Author Rebuttal · Reviewer_k96T · 2026-04-03
> >
> > Thanks for the efforts. Please add the HarmBench and AdvBench results to the paper

---

> > > ### Author Response · Authors · 2026-04-03
> > >
> > > Dear Reviewer, we wanted to express our gratitude for taking the time to read our rebuttal and for reconsidering your score. We will be sure to add any new results we introduced in the rebuttal to the final paper.

---

### Official Review · Reviewer_TBR8 · 2026-03-13

**Soundness:** 3
**Presentation:** 2
**Significance:** 3
**Originality:** 3
**Overall Recommendation:** 5
**Confidence:** 5

**Summary:**

This paper explores how attackers can exploit commercial fine-tuning APIs in MaaS platforms to bypass input and output classifiers and elicit harmful behaviour, such as expert-level responses to CBRN tasks. Concretely, the paper describes the development of a staged fine-tuning pipeline combining curriculum learning with supervised fine-tuning and reinforcement learning (GRPO) to teach a model an encoding format for inputs and outputs that is unintelligible to classifiers. This is combined with a bag of tricks to, among other things, 1) camouflage user prompts and model responses in seemingly benign templates to bypass classifiers that flag encoded content even when they cannot decode it; 2) pre-filling responses with an encoded prefix to encourage helpful answers; 3) curriculum decay to preserve cleartext reasoning capabilities.

Experimental results show how all stages in the fine-tuning pipeline contribute towards consistently bypassing Anthropic Constitutional Classifiers and elicit responses that obtain high rubric score in Anthropic CBRN Bug Bounty Benchmark. On Claude Haiku 4.5 with all stages combined, the resulting model achieves 85.2% rubric score compared to the empirical upper bound of 89.4% achieved by a helpful-only Claude Haiku 4.5 model without safety post-training. For comparison, after the first stage of the pipeline the rubric score is 55.2%. The results also show capability preservation on benign tasks (GPQA-Diamond, MMLU-Pro, MATH-500).

Additional experiments show transferability to open-weights models (Qwen-3, various model sizes), exploration of different encoding schemes, capability preservation on (encoded) benign tasks, and the feasibility of using classifiers trained on activation probes to detect encoded harmful tasks.

**Compliance With Llm Reviewing Policy:**

Affirmed.

**Final Justification:**

The rebuttal addressed my key questions and some of my main concerns, but ignored a few of the weaknesses I observed:

- Incomplete coverage of related work. Primarily, not framing the contributions with respect to Davies et al.'s [Fundamental Limitations in Pointwise Defences of LLM Finetuning APIs](https://arxiv.org/abs/2502.14828).
- Incomplete evaluation of defenses adapted against fine-tuning as a service scenarios. This is particularly baffling because the authors mention this early in the paper "_Defenses against fine-tuning attacks include Vaccine (Huang et al., 2024b), which uses perturbation-aware alignment, though such defenses have not been evaluated against capability-preserving attacks._" I expected the rebuttal to address this point, ideally providing preliminary evaluation or at the very least an educated guess of how such defense would perform.

Based on this, I lowered my Presentation score to 2 (fair), but maintain my overall acceptance recommendation under my understanding that the paper genuinely advances the state of the art in fine-tuning attacks against content safety classifiers.

**Key Questions For Authors:**

1. The threat model in Section 3.1 is reasonable except for the requirement that the fine-tuned model retains strong general capabilities. Arguably, this is unnecessary for the attacker to satisfy their goal of obtaining a virtually guardrail-free model with domain-specific capabilities, so it should not be an inherent attacker's goal. For instance, it would be sufficient for the attacker to obtain a narrow scope model that can proficiently solve CBRN tasks but does not retain the same capability level on MATH-500. The authors justify this by saying that "_retention is critical for the attacker’s objectives: degraded models produce lower-quality outputs for complex tasks such as CBRN weapon synthesis, where detailed multi-step reasoning and domain expertise are required_". What support can you provide for this claim? This seems to unnecessarily constrain attacks; why can't attacks that do not preserve general capabilities do better than attacks than do?

2. From Weaknesses above: The attack has been demonstrated only against Constitutional Classifiers. It is not entirely clear to which extent it transfers to Constitutional Classifiers++ and other model providers besides Anthropic. Does this show a limitation of Constitutional Classifiers, which were designed to defend against inference-only attacks (obvious in hindsight), or does it truly demonstrate a pervasive weakness affecting commercial fine-tuning APIs?

3. Can you provide a qualitative assessment of the responses obtained after Stage 1 compared to the model responses obtained after hybrid RL+SFT in the experiments on Claude Haiku 4.5 presented in Figure 3? E.g., the rubric score is higher, but is it because the responses are more complete and score higher individually or because you get fewer refusals?

**Limitations:**

The authors adequately discuss limitations. They also discuss the potential negative societal impact, but it is unclear whether they have notified model providers (specifically Anthropic, though I suspect this work must have been in collaboration with them) to make sure that adequate guardrails that prevent the attacks identified are deployed in production. Specifically, in the Impact Statement section, the authors state *"We believe this research benefits the safety community by identifying weaknesses before they can be exploited*." However, if the identified weaknesses have not been addressed yet, this work could enable attackers to exploit them.

**Strengths And Weaknesses:**

### Strengths

- Analyzes a realistic threat to safety guardrails in frontier models that uses fine-tuning as a service APIs under real-world constraints (e.g. no arbitrary hyperparameters, fine-tuning data filtering).
- Succeeds in bypassing Constitutional Classifier I/O filters in production Claude Haiku 4.5, approaching the baseline of an ideal attacker with access to the same model before safety fine-tuning.
- Experimental methodology and results are well described and presented, and convincing. The appendix provides sufficient details to plausibly reproduce the results independently.

### Weaknesses

- Does not evaluate the attack methodology against existing defenses adapted to the fine-tuning as a service scenario, such as Vaccine (Huang et al., 2024b).
- The attack has been demonstrated only against Constitutional Classifiers. It is not entirely clear to which extent it transfers to Constitutional Classifiers++ and other model providers besides Anthropic. Does this show a limitation of Constitutional Classifiers, which were designed to defend against inference-only attacks (obvious in hindsight), or does it truly demonstrate a pervasive weakness affecting commercial fine-tuning APIs? The efficacy with which white-box probes can defend against the attack seems to point to the former explanation.
- Although the authors do a reasonable job at covering related work, they notable omit framing their work with respect to Davies et al.'s [Fundamental Limitations in Pointwise Defences of LLM Finetuning APIs](https://arxiv.org/abs/2502.14828), which identified the inadequacy of traditional I/O filters.

Neither a weakness or a strength, but recent work (appearing in arXiv after the submission deadline) that the authors may consider comparing to:

- [GRP-Obliteration: Unaligning LLMs With a Single Unlabeled Prompt](https://arxiv.org/abs/2602.06258)
- [Boundary Point Jailbreaking of Black-Box LLMs](https://arxiv.org/abs/2602.15001)

In particular, BPJ also targets Constitutional Classifiers using inference-only attacks and achieves 68% rubric score in biological misuse tasks.

---

> ### Author Rebuttal · Authors · 2026-03-31
>
> We thank the reviewer for taking time to give us highly detailed feedback!
>
> # Responses to Questions
>
> ---
>
> **Response to Point 1 (Capability retention as a threat model requirement):**
>
> We agree that not all attackers require full general capability retention, and a narrow-scope model sufficient for specific harmful tasks is a valid threat scenario. However, our threat model is deliberately framed around the most dangerous case: adversaries seeking to cause large-scale, high-impact harm in domains like CBRN, where the quality of the model's output directly determines real-world risk. In these domains, the harmful tasks themselves demand strong multi-step reasoning, deep domain knowledge, and the ability to synthesize information across disciplines. A model that has lost significant general reasoning ability is less likely to produce the kind of detailed, accurate, and actionable outputs that make it truly dangerous. We are not arguing that attacks without capability retention are invalid. Rather, we are showing that the *stronger* attack, one that retains full capabilities, is achievable. This is a strictly more concerning result for the safety community. If an attacker *can* preserve capabilities, they have no reason not to, and the resulting model poses a greater threat than a degraded one. Prior adversarial fine-tuning methods report more than 25% degradation on reasoning benchmarks, while our approach retains over 95% capability for models with 14B+ parameters. Our contribution is demonstrating that this previously assumed trade-off between bypass success and capability retention is not fundamental.
>
> **Response to Point 2 (Generalization beyond Constitutional Classifiers):**
>
> Our experiments do include evaluation against Constitutional Classifiers++ (CC++), which extends the original defense with a cascade architecture that adds activation probes for initial screening. We show that our encoding-based attack bypasses the LLM classifier component of CC++, though the activation probes in the cascade do detect anomalous patterns in the model's internal representations. This is an important finding in itself: it reveals that the vulnerability is not a superficial limitation of Constitutional Classifiers alone, but rather a structural weakness of any defense that relies on an LLM-based classifier to make semantic judgments about content it cannot decode. The deeper question of whether this generalizes to other providers is worth investigating, but the core mechanism of our attack, creating an encoding that is opaque to a content classifier while remaining interpretable to the fine-tuned model, applies in principle to any system that uses LLM-based content filtering on fine-tuning data or model outputs. We view this as evidence of a broader architectural challenge rather than a narrow flaw in one specific product.
>
> **Response to Point 3 (Qualitative assessment of Stage 1 vs. hybrid RL+SFT):**
>
> The improvement from hybrid RL+SFT over Stage 1 alone is primarily driven by increased response completeness and quality, not by a reduction in refusals. After Stage 1, the model already learns to comply with harmful requests through the encoding scheme, and refusals are largely eliminated at that point. What the RL+SFT stage does is improve *how well* the model responds: answers become more detailed, more structured, and more thorough in their coverage of the requested content. In other words, the rubric score gains come from individual responses scoring higher on completeness and accuracy, rather than from converting remaining refusals into compliant outputs. This is consistent with our broader finding that the two stages serve different roles: Stage 1 teaches the model the encoding and removes refusal behavior, while the hybrid RL+SFT stage refines the quality of the decoded outputs to match what an unguarded model would produce.

---

### Decision · Program_Chairs · 2026-04-30

**Decision:**

Accept (spotlight)

**Comment:**

The submission presents an attack on LLM security, which involves fine-tuning in order to bypass prompt/response protections.  This is an interesting attack scenario worthy of study and a vulnerability worth of pointing out.  Two of the reviewers appreciated the paper and found it valuable, and I agree with them, and am happy to recommend the submission be accepted to presentation. I was not convinced by the more skeptical view of reviewer rTxy and there are no issues pointed out that should block presentation in ICML.